# Response of the Desertification Landscape Patterns to Spatial–Temporal Changes of Land Use: A Case Study of Salaxi in South China Karst

**Tian Shu** [1,2,3], **Kangning Xiong** [1,2,*] and **Ning Zhang** [1,2]

1    School of Karst Science, Guizhou Engineering Laboratory for Karst Desertification Control and Eco-Industry, Guizhou Normal University, Guiyang 550001, China; 17030170025@gznu.edu.cn (T.S.); 20010170532@gznu.edu.cn (N.Z.)
2    State Engineering Technology Institute for Karst Desertification Control, Guiyang 550001, China
3    Guizhou Institute of Agricultural Science and Technology Information, Guiyang 550006, China
*    Correspondence: xiongkn@gznu.edu.cn

**Abstract:** Land use change and karst desertification (KD) are interdependent. It is crucial to investigate the relationship between the KD landscape and spatial–temporal changes in land use for effective and sustainable KD management practices in karst plateau mountains. In this study, we analyzed the spatial and temporal characteristics, evolution in the pattern of land use, and KD in the Salaxi study area from 2009 to 2019, using the landscape pattern index and KD evolution trajectories, and discussed their response relationships. The results revealed the following: (1) In Salaxi, cultivated land predominantly transformed into shrubland, grassland, and woodland. The area of grassland, construction land, and garden land significantly increased, with respective increments of 379.85%, 157.14%, and 1847.81%. Conversely, the area of unutilized land decreased from 53.56 hm$^2$ to 8.55 hm$^2$, with the proportion declining from 0.62% to 0.10%. KD primarily occurs in shrubland, cultivated land, and woodland. (2) The areas of non-KD and potential KD have increased. There was a noticeable conversion of light and medium KD into potential KD, with areas of 1206.84 hm$^2$ and 459.47 hm$^2$, respectively. The KD landscape is dominated by stable and weakening ecological restoration. The comprehensive ranking of the incidence of soil KD in the study area is as follows: yellow soil > yellow-brown soil > coarse bone soil > limestone soil > purple soil. (3) The land use landscape index, the evenness index, and the fragmentation index in the demonstration area increased by 0.263, 0.120, and 0.534, respectively, while the KD landscape index, evenness index, and fragmentation index decreased by 0.360, 0.123, and 1.098, respectively. Additionally, the spreading index and aggregation index of the land use landscape decreased by 9.247 and 3.086, respectively, while the KD landscape's spreading index and aggregation index increased by 6.688 and 0.430, respectively. Both the sub-dimension indexes of the land use landscape and the KD landscape increased by 0.009. Overall, the landscape pattern of KD changes in response to land use variations and different land types exhibited varying responses to KD. The study of KD and land use landscape patterns can provide references for national strategies on KD control and the development of ecological industries.

**Keywords:** landscape pattern index; evolution trajectory; land use; karst desertification; South China Karst

## 1. Introduction

Karst landscapes are formed through the dissolution of soluble rocks by water, resulting in various landscapes and phenomena formed on the surface and underground [1]. The South China Karst, centered on the Guizhou plateau, is the largest and most concentrated, contiguous, ecologically fragile karst area among the world's three major concentrated karst distribution areas. It covers an area of over $55 \times 10^4$ km$^2$ and represents one of the

most typical and complex karst developments, with rich landscape types in the tropical–subtropical region. However, the prominent contradiction between human activities and nature in this region has led to severe ecosystem destruction, increased soil erosion, extensive land degradation, and the widespread exposure of bedrock [2–4]. One of the most significant eco-environmental problems in this context is rocky desertification [5], which refers to serious soil erosion, decline in land productivity, and land degradation processes that are similar to a desert landscape on the surface under the fragile karst plateau environment [6,7]. With the rapid increase in population and intensification of human activities, the scope and intensity of land development and utilization have expanded. The unreasonable utilization of land has led to the deterioration of ecosystems and the expansion of karst desertification [8]. Presently, the increasing degradation of KD has seriously hindered the sustainable development of the local economy and society and has become the most pressing ecological and environmental problem in southwest China.

Land use and land cover change (LUCC) is the most significant expression of the interaction between human activities and the natural environment [9]. Landscape patterns, which reflect changes in various landscape elements at a certain spatial and temporal scale, are crucial to understand land use change in landscape ecological research [10]. The fragile karst ecological environment combined with irrational human land use practices is the leading cause of KD [11,12]. This combination has accelerated landscape evolution and fragmentation in the karst mountains of southern China, characterized by "karst desertification" [13]. In order to effectively manage the ecological environment of karst areas, China has implemented various ecological construction projects, such as land consolidation, protection of natural forests and ecological public welfare forests, mountain reforestation, return of cultivated land to forests and grasses, comprehensive management of KD, improvements to sloping land, and precise poverty alleviation [14,15]. Ecological reconstruction in the karst area of southwest China has transitioned from traditional high-intensity human interference to large-scale natural restoration and artificial afforestation [16]. The area affected by KD has shifted from a continuous expansion to a net reduction, and the management of KD has progressed from effective containment to comprehensive promotion [17].

Land use structural changes are closely related to the evolution of KD, and with the advancement of comprehensive KD management, the landscape pattern of KD changes along with the changing types of land use [18]. Numerous studies have been conducted on the ecological patterns of KD and landscape resulting from land use in the karst regions. For example, Lu et al. [19] examined the ecological patterns of the landscapes in karst mountains, while Bai et al. [20] explored the landscape of karst desertification and its ecological effect. Li et al. [21] and Wang et al. [18] conducted quantitative studies on the relationship between land use and KD in the peak depressions of typical KD areas. Gao et al. [22] investigated the distribution characteristics of land use in KD areas with different landscapes, and Chen et al. [23] analyzed the correlation between land use and KD evolution under different lithologies. Ai et al. [24] accurately identified macro-scale information of KD patches and quantitatively analyzed their evolution process, providing important decision-making foundations for comprehensive KD management. Although the aforementioned studies yielded valuable insights into the relationship between land use and KD and their underlying mechanisms, most of them are qualitative studies that employed techniques, such as superposition analysis and correlation analysis. Comprehensive and in-depth studies examining the relationship between land use and KD are lacking, which hinders KD management and evaluation of the effectiveness of recent KD control projects.

Guizhou province, with its karst plateau mountains, represents the main part of the province, with a karst area that accounts for 73% of the land area, and its KD area covers 21.34% of the province's land area. KD management is the primary task and challenge in ecological restoration [25]. Therefore, this study focused on the demonstration area of integrated KD management in the highland mountainous area of Guizhou, which represents the general structure of karst environment types in southern China, namely,

the Bijie Salaxi karst plateau mountainous area (referred to as the "Salaxi Demonstration Area" hereafter) [26]. The objectives of this study are to identify the spatial and temporal evolution of land use and the landscape pattern of KD in the demonstration area and to reveal the ecological conditions and spatial variability of the karst region. The findings aim to provide a scientific basis for evaluating the effectiveness of KD management and facilitating the development of eco-industry.

## 2. Materials and Methods

### 2.1. Study Area

The Salaxi Demonstration Area is situated in the Qixingguan District of Bijie City, Guizhou Province. It is a tributary of the Liuchong River basin and spans specific geographic coordinates ranging from E105°01′10″ to 105°08′39″ and N27°11′08″ to 27°17′30″ (Figure 1). The area encompasses several villages, namely, Chaoying, Zhongshan, Chongfeng, Yongfeng, Longfeng, Shale, Shuiying, Sala, and Maoping, located in Yejiao Township of Salaxi Town. With a land area of 8627.19 hm$^2$ and a resident population of 20,215, of which the agricultural population constitutes over 98%. It exhibits characteristics of a humid monsoon climate within the northern subtropical region. The average annual temperature is 12 °C, while the average annual rainfall measures 984.40 mm. Rainfall predominantly occurs between May and September. The area's elevation ranges from 1495 to 2200 m and features a high eastern terrain that gradually transitions into a gentler western slope. The surface terrain naturally extends into hilly slopes, exposing Permian sand shale, limestone, and chert, with the predominant soil type being zoned yellow loam. The area's vegetation primarily comprises subtropical coniferous broad-leaved mixed forest and deciduous broad-leaved forest, exhibiting a vegetation coverage rate of 37.06%. Soil erosion within the area is predominantly categorized as slight to medium, with a KD area of 5593.08 hm$^2$ (Figure 2). In addition, the Salaxi Demonstration Area has undergone a long period of comprehensive rock desertification management [27]. Overall, the Salaxi Demonstration Area represents a typical karst plateau with a mountainous light–medium KD area [28].

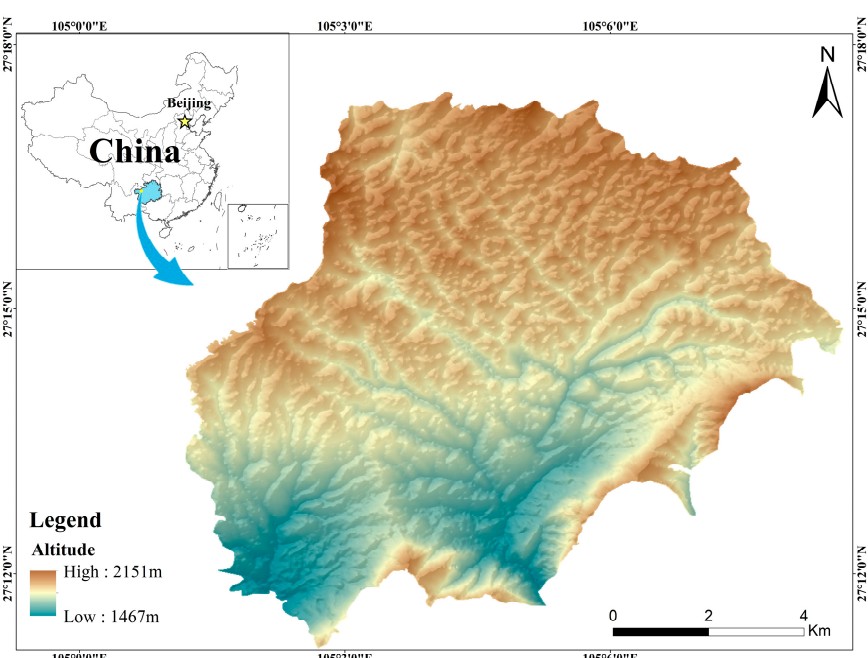

**Figure 1.** Location of the study area.

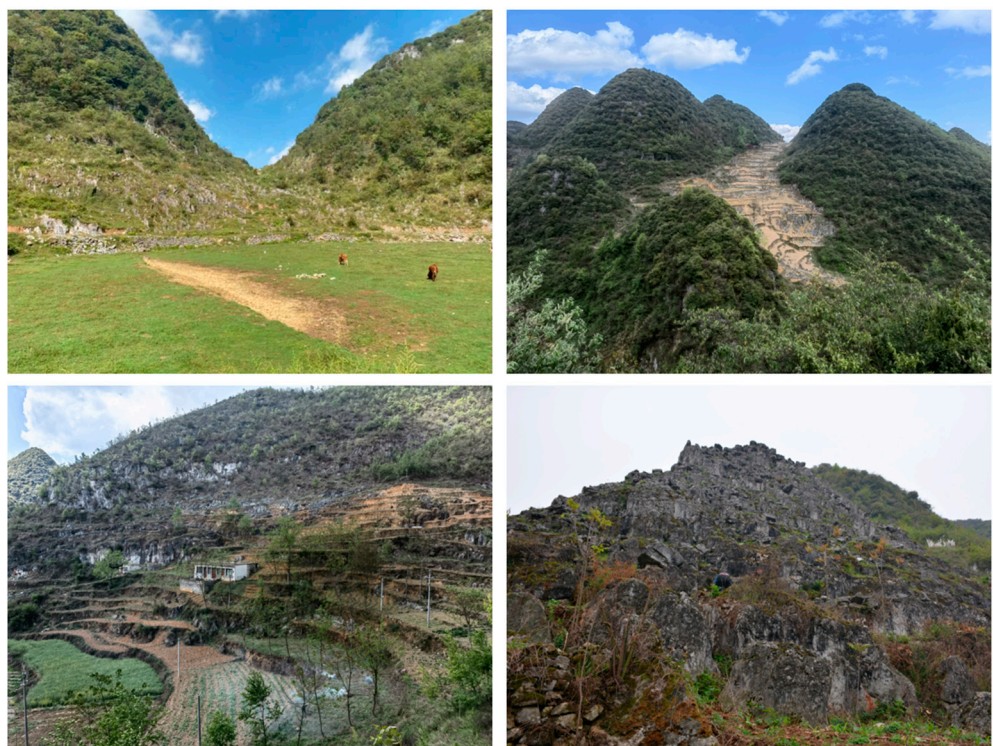

**Figure 2.** Current status of KD in the study area. (Note: These photos were taken during September 2019 in the Salaxi study area).

*2.2. Data Source and Pre-Processing*

For this study, a significant period was chosen that corresponded to China's poverty alleviation efforts and the rapid development of agriculture, forestry, and animal husbandry. Remote sensing images were acquired utilizing SPOT5 data from France (with a panchromatic band spatial resolution of 2.5 m and multispectral band spatial resolution of 10 m) on 14 April 2009 and Landsat8 OLI data on 29 March 2019. Additionally, supplementary satellite data from Google Earth, obtained on 31 October 2018, with a download accuracy of 17 levels and a spatial resolution of 2.38 m, were fused with the panchromatic and multispectral bands, respectively. This fusion process ensured a consistent spatial resolution of image data for accurate interpretation of land use types and KD grades within the demonstration area. The land use types were classified using supervised classification methods combined with field investigations. We divided the land use into nine categories (separability all greater than 1.8), and the results were validated using the Kappa coefficient (0.853). These categories included cultivated land, woodland, shrubland, open woodland, garden land, grassland, construction land, water, and unutilized land. The soil types in the demonstration area were vectorized according to the Bijie soil type map (1:50,000) after correction. To interpret the KD grades in the Salaxi research area, various factors were considered, such as topographic slope, stratigraphic lithology, bedrock exposure rate, soil thickness, and vegetation cover. Layer overlay and spatial analysis were performed using ArcGIS software(Figure 3), ultimately resulting in the generation of a comprehensive distribution map that illustrated the KD grades across the study area [29].

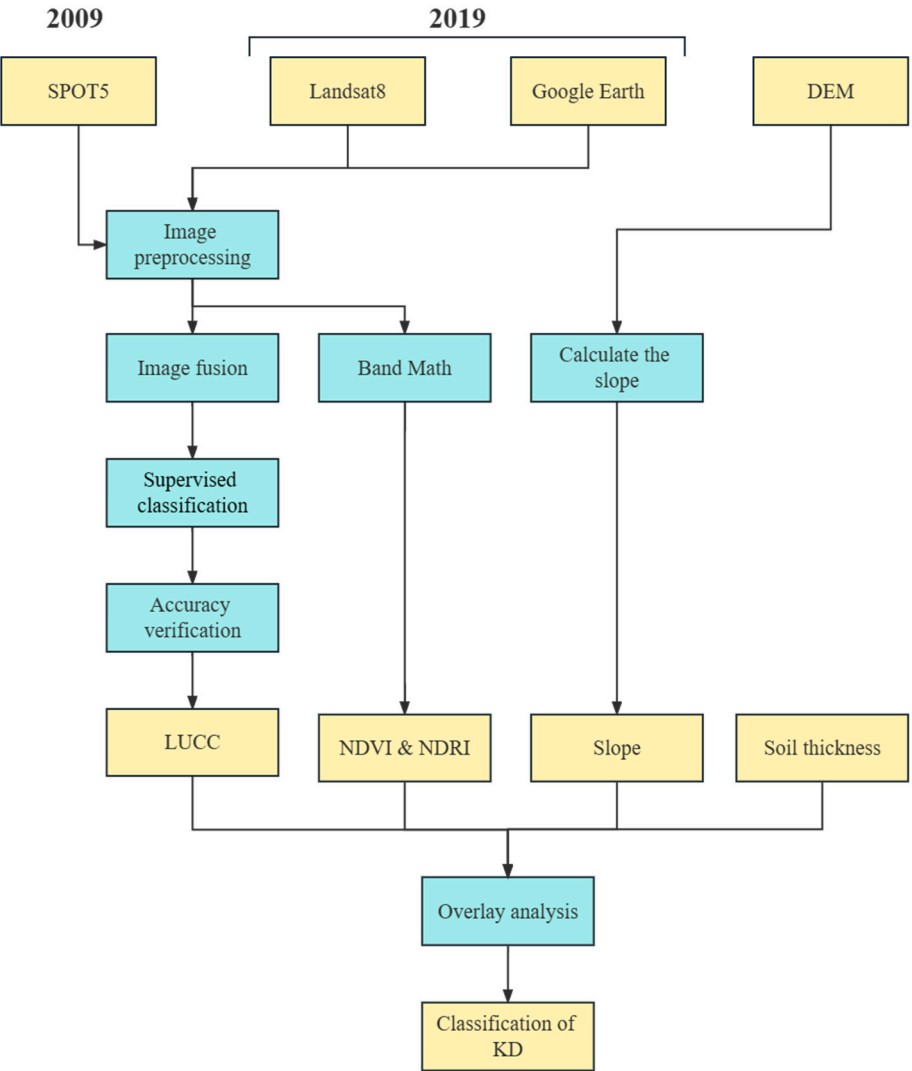

**Figure 3.** Data processing flowchart.

*2.3. Methodology*

2.3.1. KD Evolution Trajectory

The degree of change in the karst desertification (KD) data was classified into three categories: stable, weakening, and enhancing. This study focused on analyzing data from two periods, specifically 2009 and 2019. The stable type indicates no change in the level of KD from 2009 to 2019 within the spatial cell. The weakening type refers to a reduction in KD levels in 2019 compared to 2009, indicating ecological restoration. Conversely, the enhancing type signifies an aggravation of karst desertification levels in 2019 compared to 2009, indicating ecological degradation. The trajectory of KD evolution [30] spatially reflects the continuous succession process of patches within the demonstration area. By comprehensively analyzing the land use types and the distribution of KD grades in 2009 and 2019 using an ArcGIS spatial overlay, changes in the KD grade distribution among different land use types in the Salaxi Demonstration Area over a two-year period were obtained.

2.3.2. Landscape Pattern Index

The landscape pattern index serves as a comprehensive tool for condensing information related to landscape patterns. It enables the analysis of the landscape structure and function by quantitatively describing the spatial pattern and heterogeneity of the landscape, ultimately revealing the intrinsic regularities within the landscape [31]. In this study, the landscape pattern index was employed to depict the structural characteristics of land use

types and KD classes, allowing for a better understanding of the spatial and temporal changes in land use types and KD landscapes. Building upon previous research [32–35] and considering the ecological significance and research objectives, six metrics were selected: the Shannon diversity index (SHDI), the uniformity index (SEDI), the fragmentation index (FN), the spreading index (CONTAG), the fractional dimension index (FD), and the aggregation index (AI). To facilitate the analysis, the land use type and KD grade distribution data in 2009 and 2019 were converted into 5 m × 5 m raster data. Fragstats 4.2 software was utilized to analyze the land use and KD landscape patterns within the study area.

The *SHDI* reflects landscape heterogeneity and is more sensitive to the unbalanced distribution state of landscape types. The formula is as follows:

$$SHDI = -\sum_{i=1}^{m} p_i * Ln(p_i) \tag{1}$$

The *SHEI* is the diversity index under the condition of maximum homogeneity. The formula is as follows:

$$SHEI = (H/H_{\max}) * 100\% \tag{2}$$

The $C_i$ reflects the degree of fragmentation of the landscape and indicates the complexity of the spatial structure of the landscape. The formula is as follows:

$$C_i = \frac{N_i}{A_i} \tag{3}$$

The *CONTAG* describes the degree of clustering or extension trend of different patch types; the higher the value, the better the connectivity of the dominant patches of the landscape and the lower the fragmentation of the landscape. The formula is as follows:

$$CONTAG = \left[ 1 + \sum_{i=1}^{n} \sum_{j=1}^{m} p_{ij} \ln(p_{ij}) \frac{p_{ij} \ln(p_{ij})}{2 \ln(M)} \right] * 100 \tag{4}$$

The *FDI* indicates that the higher the number of sub-dimensions, the more complex the geometry of the landscape. The formula is as follows:

$$FDI = \frac{1}{2 \ln(p_i/4) / \ln A} \tag{5}$$

The *AI* reflects the connectivity between patches of landscape types; the smaller the value, the more discrete the landscape. The formula is as follows:

$$AI = \left[ \frac{g_{ii}}{\max \to g_{ii}} \right] * 100 \tag{6}$$

The index meanings are as follows: 1. *SHDI* is the Shannon diversity index, which measures the complexity of the structure; $P_i$ is the proportion of the total area of the landscape occupied by each patch type, and *m* is the total number of patch types. 2. *SHEI* is the Shannon evenness index; *H* is the diversity index, and $H_{max}$ is the maximum value of the diversity index. 3. $C_i$ is the fragmentation index; $N_i$ is the total number of patches in landscape type *i*, and $A_i$ indicates the area of landscape type *i*. 4. *CONTAG* is the spreading index; *m* is the total number of all patch types in the landscape; *n* is the number of patches in a given patch type, and $P_{ij}$ is the probability that patch types *i* and *j* are adjacent. 5. *FDI* is the sub-dimensionality index; $P_i$ is the perimeter of patch *i*, and *A* is the area of patch *i*. 6. *AI* is the aggregation index; $g_{ii}$ is the number of similar neighboring patches of the corresponding landscape type.

2.3.3. Land Use Transfer Matrix

The land use transfer matrix is a quantitative description of the transformation process and transfer status of the area between various land use elements within a period of time. It can comprehensively reflect the conversion direction of land use in a specific period of time, and its expression is given by the following:

$$Aij = \begin{pmatrix} A_{11} & \cdots & A_{1n} \\ \vdots & \ddots & \vdots \\ A_{n1} & \cdots & A_{nn} \end{pmatrix} \tag{7}$$

where *A* represents the area of each land use type, *n* represents the number of land use types, and *Aij* represents the area of land use type *i* transferred to land use type *j*.

**3. Results**

*3.1. Land Use Dynamic Change and Its Transfer Matrix*

Figure 4 displays the spatial distribution of land use types in the Salaxi Demonstration Area. Cultivated land, shrubland, and woodland emerged as the primary land use types. In 2009 and 2019, the combined area of cultivated land, shrubland, and woodland accounted for 95.52% and 83.61% of the total area of the demonstration zone, respectively. The cultivated land is predominantly dry land that is concentrated in relatively flat sections and non-karst plateau areas. The other land use types form linear strips on gentle slopes, alternating with shrubbery and woodland. Furthermore, there is a proportional increase in shrubland concurrent with a decrease in woodland. Grassland, in 2009, appeared as patchy distributions in the northern and southern regions of the demonstration area. However, by 2019, it underwent rapid expansion around shrubland, woodland, and cultivated land, encompassing an area of 154.45 hm$^2$ and 741.13 hm$^2$, reflecting a change rate of 25.70%. Notably, the area of construction land significantly increased, while unutilized land, primarily consisting of bare rocky gravel and bare land, exhibited a substantial decrease from 53.56 hm$^2$ to 8.55 hm$^2$. The proportion of unutilized land declined from 0.62% to 0.10%. Over the past decade, comprehensive karst desertification management projects, including reforestation, afforestation, conversion of cultivated land to forests and grasslands, precise poverty alleviation measures, and adjustments in agricultural industry structures, resulted in an expansion of grassland, shrubland, construction land, and garden land in the demonstration area. Conversely, cultivated land, woodland, and unutilized land experienced reductions, while other land types remained relatively stable.

The spatial overlay analysis of the distribution of land use types in the Salaxi Demonstration Area in 2009 and 2019 yielded a land-use type transfer matrix (Table 1). Among the total land transfer, the largest amounts occurred from cropland to shrubland, woodland, and construction land, amounting to 2385.25 hm$^2$, 1478.07 hm$^2$, and 630.67 hm$^2$, respectively. Notably, the most significant transfer characteristics were observed among the four major types, with cropland converting to shrubland, accounting for an area of 1213.66 hm$^2$, and cropland converting to woodland, accounting for 567.44 hm$^2$. Additionally, conversions from cropland to grassland and from woodland to shrubland were also substantial. These changes primarily stemmed from initiatives, such as reforestation, grassland restoration projects, mountain reforestation projects, and land reclamation.

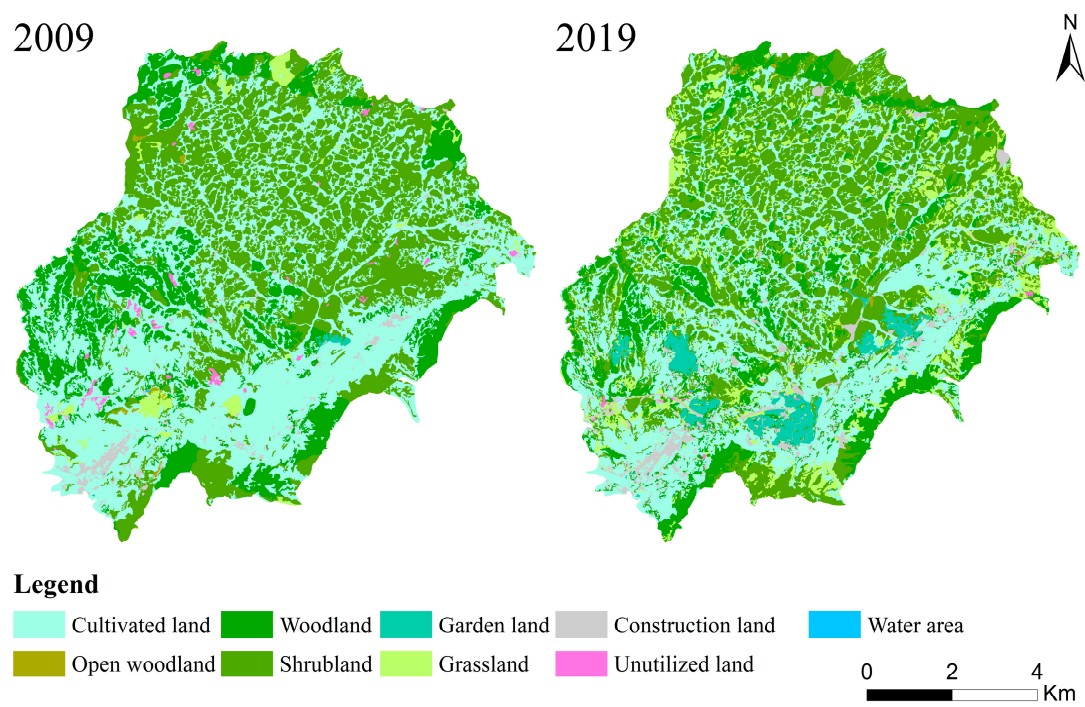

**Figure 4.** Land use and transformation types in 2009 and 2019.

**Table 1.** Land use area change table of study area from 2009 to 2019.

| Type | Grassland | Cultivated Land | Shrubland | Construction Land | Unutilized Land | Open Woodland | Water Area | Woodland | Garden Land | Sum of Transformation |
|---|---|---|---|---|---|---|---|---|---|---|
| Grassland | 13.18 | 13.27 | 40.21 | 11.35 | 0.00 | 0.09 | 0.00 | 7.31 | 23.48 | 95.71 |
| Cultivated land | 328.74 | 1375.61 | 1213.66 | 157.98 | 2.64 | 7.17 | 0.00 | 567.44 | 107.62 | 2385.25 |
| Shrubland | 281.09 | 735.68 | 1347.36 | 101.52 | 1.29 | 0.07 | 0.00 | 261.23 | 97.19 | 1478.07 |
| Construction land | 8.76 | 514.51 | 73.29 | 86.59 | 0.16 | 0.00 | 0.00 | 33.37 | 0.58 | 630.67 |
| Unutilized land | 9.32 | 24.11 | 17.26 | 1.53 | 0.54 | 0.00 | 0.00 | 11.41 | 0.00 | 63.63 |
| Open woodland | 0.84 | 5.86 | 1.56 | 0.26 | 0.31 | 0.00 | 0.00 | 63.86 | 0.00 | 72.69 |
| Water area | 0.00 | 0.00 | 0.00 | 0.00 | 0.00 | 0.00 | 0.84 | 0.00 | 0.00 | 0.00 |
| Woodland | 99.20 | 232.25 | 427.31 | 25.34 | 3.61 | 0.51 | 0.00 | 246.35 | 41.51 | 829.73 |
| Garden land | 0.00 | 0.37 | 0.00 | 0.04 | 0.00 | 0.00 | 0.00 | 0.00 | 0.56 | 0.41 |
| Sum of transformation | 741.13 | 2901.66 | 3120.65 | 384.61 | 8.55 | 7.84 | 0.84 | 1190.97 | 270.94 | 5556.16 |

*Unit: hm$^2$*

### 3.2. Dynamics of KD Grade and Its Transfer trajectory

The distribution maps of KD grades in the Salaxi Demonstration Area for 2009 and 2019 (Figure 5a,b) provide valuable insights. Overall, the main type of KD in the area was potential and light KD, covering areas of 2515.01 hm$^2$ and 1780.64 hm$^2$ in 2009 and 4342.54 hm$^2$ and 936.50 hm$^2$ in 2019, respectively. These areas accounted for 29.15%, 20.64%, and 50.34%, or 10.86% of the total area of the demonstration zone. The area without KD decreased from 1188.56 hm$^2$ in 2009 to 1062.43 hm$^2$ in 2019. Medium KD decreased from 684.01 hm$^2$ in 2009 to 64.60 hm$^2$ in 2019. In 2009, severe KD covered an area of 237.87 hm$^2$, accounting for 2.76% of the demonstration area. Notably, severe KD decreased to zero in 2019. Over the course of 10 years of comprehensive KD management, the area of potential KD increased rapidly, while the areas without KD, light KD, medium KD, and severe KD experienced significant reductions, leading to a substantial overall decline in KD severity. Changes in land use patterns influenced the KD grades in the Salaxi Demonstration Area. The expansion of forest land, grassland, and garden land coupled with the reduction in cultivated land and bare rocky gravel land, resulted in changes in the KD grades. The increase in vegetation cover limited the improved KD grades to some extent.

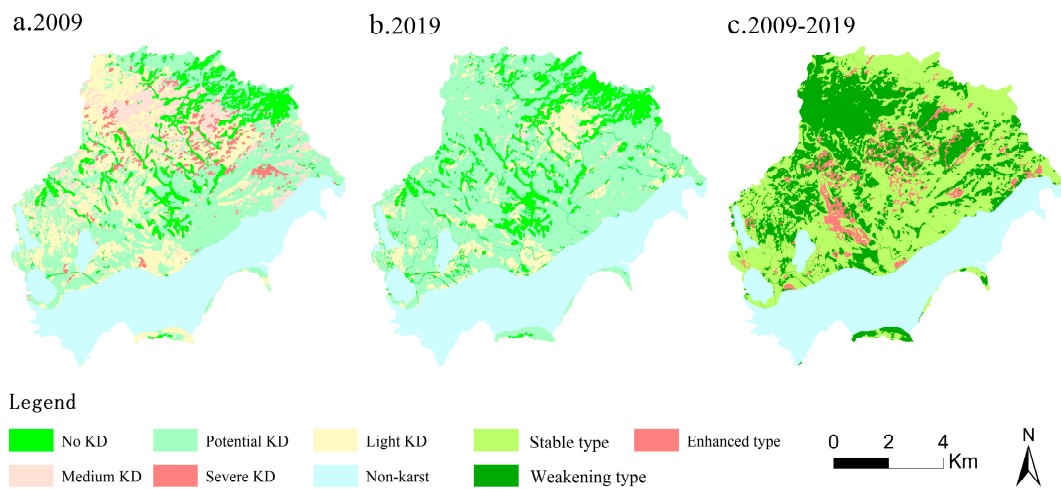

**Figure 5.** 2009–2019 Karst desertification distribution and transfer trajectory. (Note: No KD represents no karst desertification, P-KD represents potential karst desertification, L-KD represents light karst desertification, M-KD represents medium karst desertification, and S-KD represents severe karst desertification).

From the trajectory map illustrating the evolution of KD in the Salaxi research area (Figure 5c) and the data presented in Table 2, several conclusions can be drawn. Between 2009 and 2019, the area exhibiting no change in the KD grade (stable type) accounted for 3686.36 hm$^2$, representing 57.54% of the karst area in the demonstration area. The area transitioning from high-grade KD to low-grade KD (weakening type) measured 2280.65 hm$^2$, comprising 35.60% of the karst area in the demonstration area. Additionally, the area transitioning from low-grade KD to high-grade KD (enhanced type) was 437.07 hm$^2$, representing 6.85% of the karst area in the demonstration area. Notably, the enhanced type was primarily distributed in the villages of Longfeng, Zhongshan, Chaoying, and Chongfeng within the demonstration area. These findings indicate a steady recovery of KD in the Salaxi Demonstration Area, demonstrating an overall trend toward improvement with localized instances of deterioration. Furthermore, Table 2 provides additional insights. In 2019, within the landscape categories of no KD, the stable type accounted for 80.05%, and the weakening type accounted for 19.95%. Within the potential KD landscape, the stable type accounted for 52.35%, the weakening type accounted for 42.00%, and the enhancing type accounted for 5.65%. Within the light KD landscape, the stable type accounted for 55.84%, the weakening type accounted for 24.95%, and the enhancing type accounted for 19.21%. In the moderate KD landscape, the stable type accounted for 61.66%, the weakening type accounted for 17.00%, and the enhanced type accounted for 21.35% of the rock-deserted landscape. These results reveal that the landscape without KD is predominantly stable, with signs of ecological restoration. The potential, light, and medium KD landscapes exhibited a combination of the stable and weakening types, indicating ongoing ecological restoration efforts. However, there were instances of enhanced types, suggesting more challenging ecological restoration and treatment, which required long-term and appropriate prevention and control measures to avoid deterioration.

To further investigate the dynamics of the KD grades, a spatial overlay analysis was conducted using the KD grade distribution maps for 2009 and 2019 in the Salaxi Demonstration Area, generating a KD grade transfer matrix (Figure 6). In terms of total transfers, the largest amount of KD was observed in the light and medium KD categories, with transfers of 1257.69 hm$^2$ and 644.18 hm$^2$, respectively. The most significant transfer characteristics were observed between these two major grade levels, with an area of 1206.84 hm$^2$ converted from light KD to potential KD and 459.47 hm$^2$ converted from medium KD to potential KD. Additionally, transfers from no KD to potential KD and medium KD to potential KD were noteworthy. These findings indicate a shift in KD

severity from higher to lower grades, particularly in the case of severe KD transitioning to lower grades. Consequently, there has been a significant reduction in KD accompanied by an expansion of potential KD areas. However, caution should be exercised regarding the trend of no KD shifting to potential, light, and medium KD, which necessitates preventative measures. The large area covered by potential KD in the demonstration area represents a fragile point in the karst ecological environment and should be the focal point of KD prevention and control [36].

**Table 2.** The trajectory of RD transformation during 2009–2019.

| Trajectory | Year | | Area of Transformation | Percentage of Corresponding KD Degree/% | Percentage of Karst Area/% | Total |
|---|---|---|---|---|---|---|
| | 2009 | 2019 | | | | |
| Stable | No KD | No KD | 850.43 | 80.05 | 13.28 | 3686.36 |
| | P-KD | P-KD | 2273.15 | 52.35 | 35.48 | |
| | L-KD | L-KD | 522.95 | 55.84 | 8.16 | |
| | M-KD | M-KD | 39.83 | 61.66 | 0.62 | |
| Weakening | P-KD | No KD | 142.53 | 13.42 | 2.22 | 2280.65 |
| | L-KD | No KD | 49.24 | 4.63 | 0.77 | |
| | M-KD | No KD | 14.37 | 1.35 | 0.22 | |
| | S-KD | No KD | 5.86 | 0.55 | 0.09 | |
| | L-KD | P-KD | 1206.84 | 27.79 | 18.84 | |
| | M-KD | P-KD | 459.47 | 10.58 | 7.17 | |
| | S-KD | P-KD | 157.72 | 3.63 | 2.46 | |
| | M-KD | L-KD | 170.33 | 18.19 | 2.66 | |
| | S-KD | L-KD | 63.31 | 6.76 | 0.99 | |
| | S-KD | M-KD | 10.98 | 17.00 | 0.17 | |
| Enhanced | No KD | P-KD | 245.36 | 5.65 | 3.83 | 439.07 |
| | No KD | L-KD | 80.59 | 8.61 | 1.26 | |
| | No KD | M-KD | 12.18 | 18.85 | 0.19 | |
| | P-KD | L-KD | 99.33 | 10.61 | 1.55 | |
| | L-KD | M-KD | 1.61 | 2.49 | 0.03 | |
| Total | - | - | 6406.08 | - | 100.00 | 6406.08 |

Note: No KD represents no karst desertification, P-KD represents potential karst desertification, L-KD represents light karst desertification, M-KD represents medium karst desertification, and S-KD represents severe karst desertification.

### 3.3. Response Analysis of Karst Desertification and Land Use Change

3.3.1. Different KD Levels and Soil Types

To understand the changes in KD levels among different soil types in the Salaxi Demonstration Area over a two-year period, the KD rank maps and soil type maps for 2009 and 2019 were overlaid and analyzed (Figure 7 and Table 3). Given that the area covered by each soil type in the demonstration area varies, the KD incidence was utilized to assess the occurrence of KD. Table 3 provides valuable insights into the incidence of KD within the karst area of the demonstration zone for different soil types in 2009 and 2019. Notably, yellow soil exhibited the highest KD incidence, accounting for 23.36% in 2009 and 8.66% in 2019. The comprehensive ranking of KD incidence across soil types is as follows: yellow soil > yellow-brown soil > coarse bone soil > limestone soil > purple soil. Although the area covered by coarse bone soil is smaller than that of limestone soil in the demonstration area, it demonstrates a higher incidence of KD. This can be attributed to the poor soil nutrient conditions, gravel content in the topsoil, and parent material layer of

coarse bone soil. These factors contribute to the challenges associated with soil utilization and make it highly susceptible to KD [37].

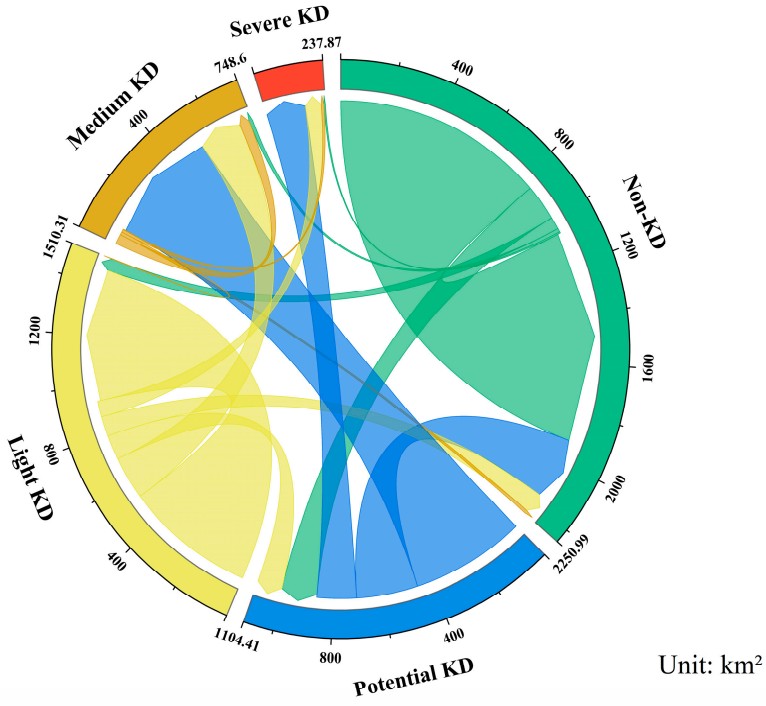

**Figure 6.** Transfer of KD degree in 2009 and 2019.

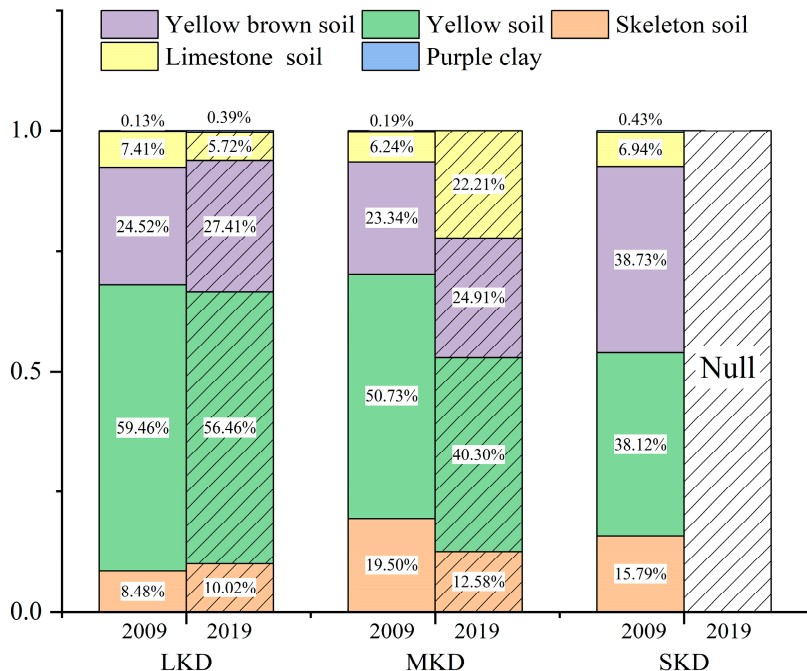

**Figure 7.** The proportion changes of KD degree under different soil types of the study area in 2009 and 2019.

**Table 3.** Different soil types and RD information in Salaxi Treatment Demonstration Area in 2009 and 2019.

| RD Degree | Year | Area of Karst/hm² | Percentage of Karst/% | Area of RD/hm² | Percentage of RD in Karst/% |
|---|---|---|---|---|---|
| Skeleton soil | 2009 | 571.04 | 8.92 | 321.99 | 5.03 |
| | 2019 | | | 101.97 | 1.59 |
| Yellow soil | 2009 | 2906.82 | 45.38 | 1496.41 | 23.36 |
| | 2019 | | | 554.79 | 8.66 |
| Yellow-brown soil | 2009 | 2099.91 | 32.78 | 688.43 | 10.75 |
| | 2019 | | | 272.78 | 4.26 |
| Lime soil | 2009 | 790.25 | 12.33 | 191.04 | 2.98 |
| | 2019 | | | 67.93 | 1.06 |
| Purple soil | 2009 | 38.05 | 0.59 | 4.63 | 0.07 |
| | 2019 | | | 3.63 | 0.06 |

### 3.3.2. Different KD Levels and Land Use

Figure 8 reveals that the KD landscape in the Salaxi Demonstration Area is primarily concentrated in cultivated land, shrubland, and forested land. In 2009, the area and proportion of mild and moderate KD in the same level were 892.61 hm² (50.13%), 607.96 hm² (34.14%), and 211.40 hm² (11.87%); and 370.20 hm² (54.12%), 287.27 hm² (42.00%), and 17.99 hm² (2.63%), respectively. In 2019, the area and proportion of mild and moderate KD in the same level were 404.18 hm² (43.16%), 359.35 hm² (38.37%), and 83.94 hm² (8.96%); and 27.66 hm² (42.81%), 22.19 hm² (34.35%), and 8.36 hm² (12.93%), respectively. In 2009, the intensity of KD was mainly concentrated in shrubland and cultivated land, with an area and proportion of 117.35 hm² (49.33%) and 108.96 hm² (45.81%), respectively. In 2019, the intensity of KD was zero. Between 2009 and 2019, moderate and intense KD showed a decreasing trend in cultivated land, shrubland, open woodland, forested land, and unutilized land. However, mild and moderate KD occurred in grasslands, but the area did not increase significantly. This also indicates that the KD management projects such as returning farmland to forest and grassland, returning to garden land, and closing mountains for forestry have played a significant role.

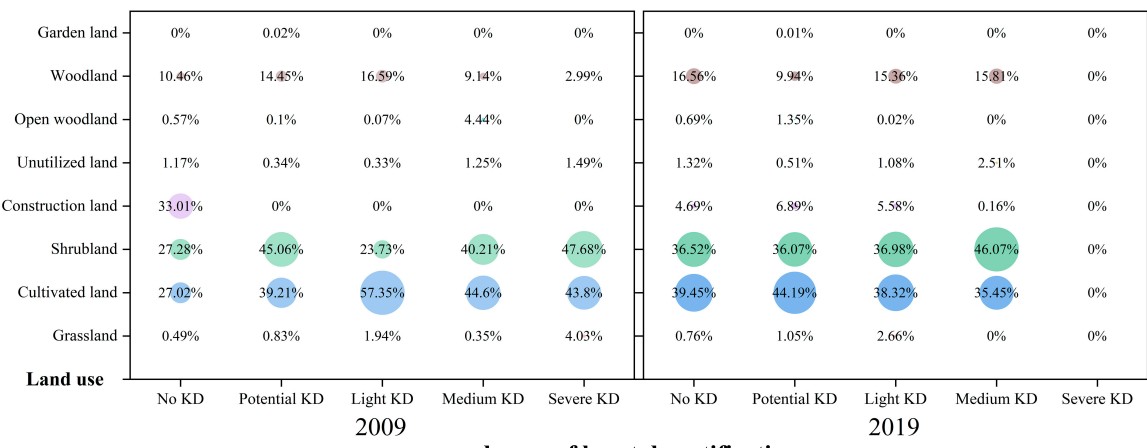

**Figure 8.** The proportion changes of some land use types under degrees of KD in the study area in 2009 and 2019.

### 3.3.3. Analysis of KD and Land Use Landscape Pattern Change

Based on the theory and method of landscape patterns, six indexes—the Shannon diversity index (SHDI), evenness index (SEDI), fragmentation index (FI), spreading index (CONTAG), fractional dimension index (FDI), and aggregation index (AI)—were selected to conduct a comprehensive analysis of land use and the KD landscape pattern in the Salaxi Demonstration Area in 2009 and 2019 (Table 4).

**Table 4.** Land use and RD landscape pattern indexes from 2009 to 2019.

| Landscape Types | Year | Shannon's Diversity Index | Shannon's Evenness Index | Fragmentation Index | Contagion Index | Fractal Dimension Index | Aggregation Index |
|---|---|---|---|---|---|---|---|
| Land use | 2009 | 1.216 | 0.554 | 0.437 | 66.839 | 1.139 | 94.865 |
| | 2019 | 1.480 | 0.673 | 0.970 | 57.592 | 1.148 | 91.780 |
| | 2009–2019 | 0.264 | 0.120 | 0.534 | −9.247 | 0.009 | −3.086 |
| KD | 2009 | 1.588 | 0.886 | 2.009 | 50.940 | 1.121 | 96.878 |
| | 2019 | 1.228 | 0.763 | 0.911 | 57.628 | 1.130 | 97.309 |
| | 2009–2019 | −0.360 | −0.123 | −1.098 | 6.688 | 0.009 | 0.430 |

The land use landscape index reveals a consistent increase in the Shannon diversity index of land use within the Salaxi Demonstration Area throughout the study period, rising from 1.216 in 2009 to 1.480 in 2019. This trend signifies that the continuous progress of KD control projects, targeted poverty alleviation, industrial structure adjustment, and urbanization have contributed to enhanced landscape diversity, heterogeneity, and complexity within the Salaxi Demonstration Area. Consequently, the differences in the areas of major land types have diminished. Furthermore, the Shannon evenness index also slightly increased, indicating that the evenness of various landscapes in the demonstration area has gradually increased, the dominant land types have decreased, and the distribution of patches in each land type has become uniform. The fragmentation index and spread index reflect the connectivity, extension, and fragmentation degree of the landscape, with an increase in the fragmentation index and a decrease in the spread index. From 2009 to 2019, the fragmentation index of the Salaxi Demonstration Zone has been increasing year by year, indicating that with the construction of infrastructure, the previously larger patches have been cut into smaller ones. The fractal dimension index is close to 1 and increased slightly in 2019 compared to 2009, indicating that the demonstration area is increasingly affected by human interference, and the landscape shape is becoming more complex. The human restoration measures taken have improved the fractal dimension index of the landscape. The degree of the convergence index decreased from 94.8652 in 2009 to 91.7795 in 2019. The KD project, the Grain for Green project, the grassland [38] project of returning cultivated land to garden land, and infrastructure construction [39] all caused large-scale landscape destruction and erosion, leading to increased landscape fragmentation, decreased spread, and poor landscape connectivity in the Salaxi Demonstration Area. In summary, human activities in the demonstration area have led to an increase in the fragmentation of land use landscapes, while also reducing the connectivity of the landscapes within the area. However, this may be beneficial for improving the ecological environment [40].

However, the changes in the KD landscape index exhibit contrasting trends compared to the land use landscape index. Throughout the study period, the KD Shannon diversity index and evenness index in the Salaxi Demonstration Area experienced a decrease. Specifically, the Shannon diversity index decreased by 0.360, indicating a reduction in the diversity of KD landscape types and a weakening in the intensity of KD. Notably, the area with intense KD reached zero in 2019, while the proportion of areas with no KD and potential KD significantly increased. Moreover, the fragmentation index decreased from 2.009 to 0.911, suggesting that with changes in land use, severe, medium, and light KD transformed into potential KD and no KD patches, leading to a gradual reduction in the overall KD area. The sub-dimensionality index increased slightly, indicating that ecological restoration efforts in the demonstration area have resulted in a slightly more complex KD landscape structure. Additionally, both the spreading index and aggregation index increased, indicating a reduction in the fragmentation of the KD landscape. Notably, the dominant patches, particularly areas with no KD, transitioned from small isolated patches to larger connected patches.

## 4. Discussion

Over the 10-year period from 2009 to 2019, significant changes in the land use types occurred in the Salaxi Demonstration Area; these were characterized by intensified degrees

and rates of change. The implementation of the comprehensive KD management project played a crucial role in these changes. The area of cultivated land experienced the most significant decrease, followed by woodland, while the reduction in unutilized land also contributed to the decline in the KD within the demonstration area. Conversely, the areas of grassland, garden land, and construction land witnessed notable increases. During the 10-year period, the intensity of KD in the demonstration area gradually diminished, and the KD grades shifted from higher grades, such as severe, medium, and light, to lower grades such as potential and no KD. Consequently, the overall KD degree decreased, and a substantial proportion of the landscape exhibited stable and weakening KD conditions, indicating an improvement in the KD situation. These positive outcomes were attributed to the comprehensive management of KD and the implementation of ecological poverty alleviation projects, which aimed to restore the ecological environment while adjusting the industrial structure. Notably, these findings align with the research results of other scholars [41]. Regarding the higher incidence of KD in yellow soil and yellow-brown soil, it is recommended to select suitable ecological industries that align with the characteristics of these soil types. This approach would ensure efficient KD management, while preserving soil quality and fertility. However, due to the limitations of the available data, the quantitative analysis conducted using two phases of images with time series compensation was insufficient to reveal the underlying mechanisms of these effects in depth. Therefore, further research is needed to explore the correlation characteristics between typical KD and land use, to update the classification of KD, and to elucidate the underlying mechanisms of these interactions.

At the same time, there has been an increase in the diversity and heterogeneity of the land use landscape in the research area. The expansion of construction land, which includes urban residential, industrial, mining, and transportation areas, has resulted in the destruction of extensive areas of arable land, grassland, and forested landscapes. Consequently, this has led to an increase in landscape fragmentation, a decrease in spread, and a deterioration in landscape connectivity within the area. The diversity of KD landscape types has been reduced, and the intensity of KD has weakened, but the ecological restoration of the demonstration area makes the structure of the KD landscape slightly complicated. Therefore, in the subsequent ecological management of KD and the project of returning farmland to forests and promoting fruit cultivation, while ensuring an increase in the area of characteristic economic forests and fruit forests, it is crucial to consider the integrated impact of other human activities on the ecological landscape pattern. This approach aims to reduce disturbances to the ecological landscape pattern, promote human welfare, foster the healthy development of the ecosystem, and serve as a valuable reference for KD management in the southwest karst region.

## 5. Conclusions

We chose the Salaxi KD comprehensive management demonstration area for this research, and the study period was 2009–2019. Based on the landscape pattern index, land use transfer, and KD evolution trajectory, we investigated the response relationship between land use and KD and drew the following conclusions:

(1) Cultivated land, woodland, and shrubland were the dominant land types in the demonstration area. Grassland, shrubland, construction land, and garden land increased over the study period, while unutilized land, such as cultivated land, woodland, bare rocky oak land, and bare land, decreased. Other land types remained stable. The obvious transformations were arable land into shrubland, woodland, and grassland and construction land into arable land.

(2) The evolution from light and medium KD to potential KD was evident. The predominance of stable and weakened KD classes indicated stabilization and ecological restoration of the KD landscape. The incidence of KD in different soil types is ranked as follows: yellow soil > yellow-brown soil > coarse bone soil > limestone soil > purple soil. Studying the relationship between soil types and KD is a future research direction.

(3) The response of the KD landscape patterns to land use changes was evident, with different land types showing variation. The occurrence of medium and severe KD in cultivated land, shrubland, open woodland, woodland, and unutilized land decreased. However, there was an increase in light and medium KD in grassland, although the overall area increase was minimal. This indicates that KD management projects, such as returning farmland to forests, grass, orchards, and gardens, as well as mountain reforestation, played a significant role in promoting these processes.

(4) The heterogeneity of landscape land use increased, as did landscape fragmentation. However, the fragmentation of the KD landscape decreased, while the connectivity of dominant patches improved. Currently, our study is restricted to the analysis of each index and does not investigate the mechanisms of action.

**Author Contributions:** Conceptualization, K.X. and T.S.; methodology, T.S.; software, T.S.; formal analysis, T.S.; resources, T.S.; data curation, N.Z. and T.S.; writing—original draft preparation, N.Z.; writing—review and editing, N.Z., K.X. and T.S.; visualization, N.Z.; supervision, K.X.; project administration, K.X.; funding acquisition, K.X. All authors have read and agreed to the published version of the manuscript.

**Funding:** This study was supported by the Philosophy and Social Science Planning Key Project of Guizhou Province, China (21GZZB43), the Key Project of Science and Technology Program of Guizhou Province (No. 5411 2017 QKHPTRC), the China Overseas Expertise Introduction Program for Discipline Innovation (No. D17016), and the Guizhou Provincial Basic Research Program (Qianhe Foundation-ZK (2021) General 130).

**Institutional Review Board Statement:** Not applicable.

**Informed Consent Statement:** Not applicable.

**Data Availability Statement:** Not applicable.

**Conflicts of Interest:** The authors declare no conflict of interest.

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
