# Peer review of "Response of the Desertification Landscape Patterns to Spatial–Temporal Changes of Land Use: A Case Study of Salaxi in South China Karst"

_land, doi:10.3390/land12081557_

Round 1

Reviewer 1 Report

This manuscript presents a significant contribution to  analysis to analyzes the response relationship between KD landscape pattern and spatial and temporal changes of land use type. However, some aspects of this work could be still improved:

- Justify the choice of the period under analysis.

- One particular issue needs addressing is placing the results of this study into the land use policy context.

- Present the theoretical, practical and political contributions of the study.

- Highlight the main limitations of the study.

 For more detailed comments, please see the attachment.

Author Response

Response to Reviewer 1 Comments

Dear Reviewer:

Thank you very much for the hard work in reviewing this paper and for allowing us to revise the manuscript. These comments have been very helpful in revising and improving the paper, and have also provided us with important guidance for our research. We have carefully studied the review comments and have made comprehensive changes and improvements in response to them. A cross-reference of the comments is shown below:

Point 1: The significance of the research topic should be explained in more detail in the introduction. The research gap should be specified in detail.

Response 1: Thank you very much for your suggestion. In the introduction, the significance of the research topic was explained in detail. Revised lines 100-107 in the manuscript as follows:

“... in the highland mountainous area of Guizhou, which represents the general structure of karst environment types in southern China, namely, the Bijie Salaxi karst plateau mountainous area (referred to as the "Salaxi demonstration area" hereafter)[27]. The objectives of this study are to identify the spatial and temporal evolution of land use and the landscape pattern of KD in the demonstration area and to reveal the ecological conditions and spatial variability of the karst region. The findings aim to provide a scientific basis for evaluating the effectiveness of KD management and facilitating the development of eco-industry.”

Point 2: In the description of the study area, could you provide more details about the type of vegetation cover? The description of the types of vegetation is very important as it directly influences the expansion of areas in the process of desertification.

Response 2: We fully agree with your suggestion. In the description of the study area, we have added a description of vegetation. Revised lines 123-128 in the manuscript as follows:

“... The area's vegetation primarily comprises subtropical coniferous broad-leaved mixed forest and deciduous broad-leaved forest, exhibiting a vegetation coverage rate of 37.06%. Soil erosion within  the area is predominantly categorized as slight to medium, with a KD area of 5593.08 hm2(Fig. 2). Overall, the Salaxi Demonstration area represents a typical karst plateau mountainous light-medium KD area [28].”

Point 3: Line 153: Please describe the reason for choosing this analysis period. Was it a period of significant urban expansion in the region or agricultural expansion?

Response 3: Thank you for your feedback. Based on your feedback, we have added the reasons for choosing this analysis period in the article. Revised lines 134-136 in the manuscript as follows:

“...For this study, a significant period was chosen that corresponds to China's poverty alleviation efforts and the rapid development of agriculture, forestry, and animal husbandry. Remote sensing images were acquired, utilizing SPOT5...”

Point 4: Please check Figure 3, as the 2009a map seems to be missing.

Response 4: Thank you very much for your suggestion. We have remade this image to make it more intuitive and clear. Revised lines 315-319 in the manuscript as follows:

Figure 5. 2009-2019 Karst desertification distribution and transfer trajectory(Note: No KD represents no karst desertification, P-KD represents potential karst desertification, L-KD represents light karst desertification, M-KD represents medium karst desertification, and S-KD represents severe karst desertification.)

Point 5: In line 273, replace the reference to Table 4 with Table 2. The same correction should be made in line 297.

Response 5: Thank you very much for your suggestion. We have carefully checked and made modifications to the table and corresponding references in the text. Revised lines 289-290 in the manuscript as follows:

“...From the trajectory map of KD evolution in the Salaxi demonstration area (Fig. 5) and Table 2, ...”

Table 2. The trajectory of RD transformation during 2009-2019

 Trajectory

Year

Area of transformation

Percentage of corresponding KD degree /%

Percentage of karst area /%

Total

2009

2019

Stable

No KD

No KD

850.43

80.05

13.28

3686.36

P-KD

P-KD

2273.15

52.35

35.48

L-KD

L-KD

522.95

55.84

8.16

M-KD

M-KD

39.83

61.66

0.62

Attenuated

P-KD

No KD

142.53

13.42

2.22

2280.65

L-KD

No KD

49.24

4.63

0.77

M-KD

No KD

14.37

1.35

0.22

S-KD

No KD

5.86

0.55

0.09

L-KD

P-KD

1206.84

27.79

18.84

M-KD

P-KD

459.47

10.58

7.17

S-KD

P-KD

157.72

3.63

2.46

M-KD

L-KD

170.33

18.19

2.66

S-KD

L-KD

63.31

6.76

0.99

S-KD

M-KD

10.98

17.00

0.17

Enhanced

No KD

P-KD

245.36

5.65

3.83

439.07

No KD

L-KD

80.59

8.61

1.26

No KD

M-KD

12.18

18.85

0.19

P-KD

L-KD

99.33

10.61

1.55

L-KD

M-KD

1.61

2.49

0.03

Total

-

-

6406.08

-

100.00

6406.08

Note: No KD represents no karst desertification, P-KD represents potential karst desertification, L-KD represents light karst desertification, M-KD represents medium karst desertification, and S-KD represents severe karst desertification.

Point 6: Replace the reference to Table 5 with Table 3.

Response 6: Thank you very much for your suggestion. We have carefully checked and made modifications to the table and corresponding references in the text. Revised lines 326-327 in the manuscript as follows:

“...the KD rank maps and soil type maps for 2009 and 2019 were overlaid and analyzed (Fig. 6 and Table 3).”

Point 7: In line 318, replace the reference to Table 5 with Table 3.

Response 7: Thank you very much for your suggestion. We have carefully checked and made modifications to the table and corresponding references in the text. Revised lines 342-343 in the manuscript as follows:

Table 3. different soil types and RD information in Salaxi Treatment Demonstration Area in 2009 and 2019

RD degree

Year

Area of karst/hm2

Percentage of karst/%

 Area of RD/hm2

Percentage of RD in karst/% 

Skeleton soil

2009

571.04

8.92

321.99

5.03

2019

 101.97

1.59

Yellow soil

2009

2906.82

45.38

 1496.41

23.36

2019

554.79

8.66

Yellow-brown soil

2009

2099.91

32.78

688.43

10.75

2019

 272.78

4.26

Lime soil

2009

790.25

12.33

 191.04

2.98

2019

 67.93

1.06

Purple soil

2009

38.05

0.59

 4.63

0.07

2019

 3.63

0.06

Point 8: In lines 322-324, the paragraph "The land use types and KD grade distribution maps in 2009 and 2019 were comprehensively analyzed using ArcGIS spatial overlay to obtain the changes in KD grade distribution of different land use types in the Salaxi demonstration area in two years (Fig. 5)" could be transferred to section 2.3.

Response 8: Thank you very much for your suggestion. After careful review of the article, we agree with your suggestion to move this section to section 2.3. Revised lines 165-169 in the manuscript as follows:

“...The trajectory of KD evolution [30] spatially reflects the continuous succession process of patches within the demonstration area. By comprehensively analyzing the land use types and distribution of KD grades in 2009 and 2019 using ArcGIS spatial overlay, the changes in KD grade distribution among different land use types in the Salaxi demonstration area over a two-year period were obtained. “

Point 9: In lines 348-349, replace the reference to Table 6 with Table 4.

Response 9: Thank you very much for your suggestion. We have carefully checked and made modifications to the table and corresponding references in the text. Revised lines 365-370 in the manuscript as follows:

...Based on the theory and method of landscape pattern, six indexes, including the Shannon diversity index (SHDI), evenness index (SEDI), fragmentation index (FI), spreading index (CONTAG), fractional dimension index (FDI), and aggregation index (AI), were selected to conduct a comprehensive analysis of the land use and KD landscape pattern in the Salaxidemonstration area in 2009 and 2019 (Table 4).”

Table 4. Land use and RD landscape pattern indexes from 2009 to 2019.

Landscape types

Year

Shannon's diversity index

Shannon's evenness Index

fragmentation index

contagion Index

fractal dimension index

aggregation index

Land use

2009

1.216

0.554

0.437

66.839

1.139

94.865

2019

1.480

0.673

0.970

57.592

1.148

91.780

2009-2019

0.264

0.120

0.534

-9.247

0.009

-3.086

KD

2009

1.588

0.886

2.009

50.940

1.121

96.878

2019

1.228

0.763

0.911

57.628

1.130

97.309

2009-2019

-0.360

-0.123

-1.098

6.688

0.009

0.430

Point 10: The conclusions could be strengthened by including evidence from the results of this study within the context of land use policies. How that areas that have increased or decreased can affect the politics of the region.

Response 10: Thank you very much for your suggestion. We have written policies on land use and rocky desertification control in the article. Revised lines 66-75 in the manuscript as follows:

“...In order to effectively manage the ecological environment of karst areas, China has implemented various ecological construction projects, such as land consolidation, protection of natural forests and ecological public welfare forests, mountain reforestation, return of cultivated land to forests and grasses, comprehensive management of KD, improvement of sloping land, and precise poverty alleviation [14][15]. Ecological reconstruction in the karst area of southwest China has transitioned  from traditional high-intensity human interference to large-scale natural restoration and artificial afforestation [16]. The area affected by KD has shifted from a continuous expansion to a net reduction, and the management of KD has progressed from effective containment to comprehensive promotion [17].”

Point 11: I would suggest adding field photos and identifying desertification areas in the study region (if possible). A discussion considering field photos is more interesting and would help to validate your results.

Response 11: Thank you very much for your suggestion. We have added on-site photos in the article, hoping to showcase the rocky desertification situation in the area. Revised lines 131-132 in the manuscript as follows:

Figure 2. Current status of KD in the study area.

Point 12: Highlight the main limitations of the study.

Response 12: Thank you very much for your suggestion. In the fourth part of the article, we emphasized the main shortcomings of the research and the prospects for the future. Revised lines  432-437 and 446-453 in the manuscript as follows:  

“... However, due to the limitations of the available data, the quantitative analysis conducted using two phases of images with time series compensation is insufficient to reveal the underlying mechanisms of these effects in depth. Therefore, further research is needed to explore the correlation characteristics between typical KD and land use, update the classification of KD, and elucidate the underlying mechanisms of these interactions.”

“...Therefore, in the subsequent ecological management of KD and the project of returning farmland to forests and promoting fruit cultivation, while ensuring the increase of the area of characteristic economic forests and warp and fruit forests, it is crucial to consider the integrated impact of other human activities on the ecological landscape pattern. This approach aims to reduce disturbances to the ecological landscape pattern, promote human welfare, foster the healthy development of the ecosystem, and serve as a valuable reference for KD management in the southwest karst region.”

Reviewer 2 Report

Thank you for considering me to review this manuscript, “Response of the desertification landscape patterns to space-temporal changes of land use: a case study of Salaxi in South China Karst. An in-depth analysis of the characteristics of KD in relationship with land use types helps formulate national land policies tailored to local conditions. However, the article still has the following problems. (Kindly check the report). 

Best regards

English is not smooth and should improve it. You use long sentences that make your idea unclear. 

Author Response

Response to Reviewer 2 Comments

Dear Reviewer:

Thank you very much for the hard work in reviewing this paper and for allowing us to revise the manuscript. These comments have been very helpful in revising and improving the paper, and have also provided us with important guidance for our research. We have carefully studied the review comments and have made comprehensive changes and improvements in response to them. A cross-reference of the comments is shown below:

Abstract

Point 1: The objective of the research has not been written clearly in this section.

Response 1: Thank you very much for your suggestion. Thank you for your feedback. The purpose of this study is to explore the response relationship between the rocky desertification landscape and the spatio-temporal change of land use, which has important reference significance for the comprehensive control of rocky desertification in Karst Plateau plateau mountains and its effects and regional sustainable development. Has been written in this section.

Point 2: Lines 15 to 17: This section's methodology explanation is confusing and must be rewritten clearly.  

Response 2: Thank you very much for your hard work. We fully agree with your opinion and have made modifications based on it. The method used has been rewritten in this section. Revised lines 13-16 in the manuscript as follows:

“... This study analyzes the landscape pattern of land use and KD in the Salaxi research area from 2009 to 2019 using remote sensing and GIS spatial analysis technology. The detailed changes in land use and landscape pattern are discussed. “

Point 3: Line 18-19, “The spatial and temporal changes were analyzed comprehensively,” may be unnecessary, or reconsider this sentence. The mistake is noted in line 24 “The area of no KD and potential KD increased.”

Response 3: We fully agree with your opinion and have deleted the phrase 'comprehensive analysis of spatial and temporal changes'.

Point 4: Add the study's main policy implications and limits in the abstract section.

Response 4: Thank you for your feedback. We have rewritten the abstract section to add the main significance of the research. Revised lines 32-34 in the manuscript as follows:

“... The study of KD and land use landscape patterns can provide references for national strategies on KD control and the development of ecological industries.”

Point 5: Globally the abstract need to be rewritten concisely and clearly.

Response 5: Thank you very much for your hard work. We have reorganized the abstract. Revised lines 11-34 in the manuscript as follows:

“Land use change and karst desertification (KD) are interdependent. It is crucial to investigate the relationship between the KD landscape and spatial-temporal changes in land use for effective and sustainable KD management in karst plateau mountains. This study analyzes the landscape pattern of land use and KD in the Salaxi research area from 2009 to 2019 using remote sensing and GIS spatial analysis technology. The detailed changes in land use and landscape pattern are discussed. The results reveal the following: (1) In Salaxi, cultivated land has predominantly transformed into shrubland, grassland, and woodland. The area of grassland, construction land, and garden land has significantly increased, with respective increments of 379.85%, 157.14%, and 1847.81%. Conversely, the area of unutilized land has decreased from 53.56 hm2 to 8.55 hm2, with the proportion declining from 0.62% to 0.10%. KD primarily occurs in shrubland, cultivated land, and woodland. (2) The area of non-KD and potential KD has increased. There is a noticeable conversion of light and medium KD into potential KD, with areas of 1206.84 hm2 and 459.47 hm2, respectively. KD landscape is dominated by stable and attenuated ecological restoration. The comprehensive ranking of soil KD incidence in the study area is as follows: yellow soil > yellow-brown soil > coarse bone soil > limestone soil > purple soil. (3) The land use landscape index, evenness index, and fragmentation index in the demonstration area have increased by 0.263, 0.120, and 0.534, respectively, while the KD landscape index has decreased by 0.360, 0.123, and 1.098. Additionally, the spreading index and aggregation index of land use landscape have decreased by 9.247 and 3.086, respectively, while the KD landscape index has increased by 6.688 and 0.430. Both the sub-dimension index of land use landscape and KD landscape have increased by 0.009. Overall, the landscape pattern of KD changes in response to land use variations, and different land types exhibit varying responses to KD. The study of KD and land use landscape patterns can provide references for national strategies on KD control and the development of ecological industries.”

Introduction

Point 1: The preface introduction is too weak to summarize previous research, and some newest related literature should be added.

Response 1: Thank you very much for your hard work. We have summarized previous research in the introduction section and added the latest relevant literature. Revised lines 80-88 in the manuscript as follows:

“...Lu et. al [19] examined the ecological patterns of the landscapes in the Karst mountains, while Bai et. al [20] explored the  landscape of karst desertification and its ecological effect. Li et. al [21] and Wang et. al [18] conducted quantitative studies on the relationship between land use and KD in the peak depressions of typical KD areas. Gao et. al [22] investigated the distribution characteristics of land use in KD areas with different landscapes, and Chen et. al [23] analyzed the correlation between land use and KD evolution under different lithologies. Ai et. al [24] accurately identified macro-scale information of KD patches and quantitatively analyzed their evolution process, providing important decision-making foundations for comprehensive KD management.”

Point 2: The connotation of karst desertification(KD) must be explained in the context of this investigation.

Response 2: Thank you for your feedback. We have indicated the connotation of rocky desertification in the text. Revised lines 49-52 in the manuscript as follows:

“...One of the most significant ecological environment problems in this context is rocky desertification [5], which refers to the serious soil erosion, decline in land productivity, and land degradation process similar to a desert landscape on the surface under the fragile Karst Plateau environment [6]-[7]. ”

Point 3: In addition, you must explain the relationship between land use change, karst desertification and its mechanism.

Response 3: Thank you for your feedback. We have indicated the connection between the three in the text. Revised lines 63-66 in the manuscript as follows:

“...The fragile karst ecological environment combined with irrational human land use practices is the leading cause of KD [11][12]. This combination has accelerated landscape evolution and fragmentation in the karst mountains of southern China, characterized by "karst desertification" [13]. ”

Point 4: Lines 99 to 112: This section is confusing: For instance, lines 102 to 106; make clear the study's objective etc.

Response 4: Thank you for your hard work. In this section, we introduced the background of the research area and the significance of doing this work.

Point 5 and 6 and 7: The introduction part should include why this study is essential; What is missing? What is needed?

Response 5 and 6 and 7: Thank you for your hard work. In the last paragraph of the introduction, we introduced the necessity of this study. Revised lines 96-107 in the manuscript as follows:

“Guizhou province, with its karst plateau mountains, represents the main part of the province, with karst area accounts for 73% of the land area, and KD area covering 21.34% of the province's land area. KD management is the primary task and challenge in ecological restoration [25]-[26]. Therefore, this study focuses on the demonstration area of integrated KD management in the highland mountainous area of Guizhou, which represents the general structure of karst environment types in southern China, namely, the Bijie Salaxi karst plateau mountainous area (referred to as the "Salaxi demonstration area" hereafter)[27]. The objectives of this study are to identify the spatial and temporal evolution of land use and the landscape pattern of KD in the demonstration area and to reveal the ecological conditions and spatial variability of the karst region. The findings aim to provide a scientific basis for evaluating the effectiveness of KD management and facilitating the development of eco-industry.”

Materials

Point 1: Lines 119-120 “…..total population of 20215…” maybe after 20215, you should add inhabitants because the connotation of “population” is not only concerned humans.

Response 1: Thank you very much for your feedback. We fully agree with your proposal and have made modifications in the manuscript. Revised lines 115-116 in the manuscript as follows:

“... With a land area of 8627.19 hm2 and a resident population of 20215, of which agricultural population constitutes over 98%.”

Point 2: Lines 122-123 “…..with rainfall mainly concentrated in May-September” Concentred or extended from May-September? Kindly check it.

Response 2: Thank you for your hard work. The rainfall mainly starts in May and ends in September. We have made modifications to this point in the article. Revised lines 118-119 in the manuscript as follows:

“...Rainfall predominantly occurs between May and September.”

Point 3: Line 123 “The elevation is 1495-2200 m.” What is meaning? The same “the terrain or topography?

Response 3: Thank you very much for your feedback. The meaning of this paragraph in the manuscript is that the altitude of the research area is between 1495 meters and 2200 meters. And language modifications were made in the manuscript. Revised lines 119-120 in the manuscript as follows:

“...The area's elevation ranges from 1495 to 2200 meters,...”

Point 4: In the study area presentation, it will be very interesting to add the current status of karst desertification(KD).

Response 4: Thank you very much for your feedback. We strongly agree with your opinion. In the introduction of the research area, we have added the current situation of rocky desertification and on-site photos. Revised lines 123-132 in the manuscript as follows:

“... The area's vegetation primarily comprises subtropical coniferous broad-leaved mixed forest and deciduous broad-leaved forest, exhibiting a vegetation coverage rate of 37.06%. Soil erosion within  the area is predominantly categorized as slight to medium, with a KD area of 5593.08 hm2(Fig. 2). Overall, the Salaxi Demonstration area represents a typical karst plateau mountainous light-medium KD area [28].”

Figure 2. Current status of KD of the study area.

Point 5: Lines 140-142: You classified the land use types into nine classes that correspond to level 1, maybe; what are the classification types at level 2?

Response 5: Thank you very much for your feedback. In response to your question, we will provide the following answers. We classify land use into nine categories without a second classification. In the future work, we will pay attention to this point, and select Satellite imagery with higher accuracy to make the classification of land use more precise.

Point 6: Lines 143: ….” combined with the actual situation of the demonstration area…” What is the meaning of this sentence?

Response 6: Thank you for your hard work. The meaning of this sentence is that while conducting remote sensing interpretation, we also conducted on-site research. In order to avoid misunderstandings between you and the readers, we have made modifications to this sentence. Revised lines 145-149 in the manuscript as follows:

“...The land use types were classified using supervised classification methods combined with field investigations. Nine distinct categories were identified, with the classification accuracy assessed using the Kappa coefficient. These categories include cultivated land, woodland, shrubland, open woodland, garden land, grassland, construction land, water, and unutilized land. ”

Point 7: Line 144” KD level was divided regarding the classification of KD grade” What is meaning?

Response 7: Thank you for your hard work. The classification of KD levels refers to the classification of rocky desertification based on our team's previous understanding of remote sensing images and other factors, and the classification of KD levels based on the interpretation results, just like the classification of vegetation coverage.

Point 8: Lines 146 to 149 ……”main reading factors…” How and what is the use of these data?

Response 8: Thank you very much for your feedback. This is our problem, and we humbly accept it. We have made revisions to the language of this sentence in the manuscript. It should be the main interpretation factor. Revised lines 151-155 in the manuscript as follows:

“...and vegetation cover. Supplementary information on average rainfall, soil erosion, and land use in the area was also incorporated. Layer overlay and spatial analysis were performed using  ArcGIS software, ultimately resulting in the generation of a comprehensive distribution map illustrating the KD grades across the study area [29].”

Point 9: You use the series time from 2009 and 2019; why do you ignore the time from 2019 to 2022, for example? If you ignore this period, is it interesting to provide suitable policy implications?

Response 9: Thank you for your hard work. The reason why we chose this period is that it is a critical period for China's poverty alleviation efforts, and the development of agriculture, forestry, and animal husbandry is rapid. Choosing this stage can better illustrate the effectiveness of rocky desertification control. Revised lines 134-136 in the manuscript as follows:

“...For this study, a significant period was chosen that corresponds to China's poverty alleviation efforts and the rapid development of agriculture, forestry, and animal husbandry. “

Point 10 and 11: The method of land use classification is unclear; is it a supervised or unsupervised classification? If it is supervised, what is the classifier, and what are the parameters? What is the reliability/accuracy of the results?

Response 10 and 11: Thank you very much for your feedback. The classification method for land use is supervised classification and we conducted on-site investigations. The Kappa coefficient also passed the consistency test.

Point 12: The processing of satellite data is unclear in each pre-processing step. For example, in line 140, “after correction,” Which? Please, you can describe the process.

Response 12: Thank you very much for your feedback. We conducted preprocessing work on satellite data, including data fusion, atmospheric correction, radiometric calibration, and data cropping. The Spatial analysis function of ArcGIS is used to process various data.

Point 13: Different remote sensing data have different spatial resolutions; how have you solved this matter?

Response 13: Thank you very much for your feedback. Facing remote sensing data with different spatial resolutions, we first performed geometric correction and spatial registration. Finally, Image fusion is performed for images with different spatial resolutions.

Point 14: The subsection of 2.3. is not correct. There are two points 2.3. Kindly Check it—also, in the text lines 152 to 158. What would you aim to describe in this section?

Response 14: Thank you very much for your feedback. This is our mistake, please understand. We have modified the issue with the title number. In this section, we are laying the groundwork for the analysis of the results in section 3.2.

Point 15: Line 170, which table 1?

Response 15: Thank you very much for your feedback. This is our mistake, please understand. We have modified the serial number of each table and checked it throughout the entire text.

Point 16: Line 179 what is the significance of FI?

Response 16: Thank you very much for your feedback. We have changed FI to Ci. I hope to express the meaning of the formula more clearly. Revised lines 191-192 in the manuscript as follows:

“...Ci reflects the degree of fragmentation of the landscape and indicates the complexity of the spatial structure of the landscape. The formula is:”

Point 17: It's a suggestion to describe the equation's significance after each equation. The Index meanings described in the section from lines 192 to 201 are confusing.

Response 17: Thank you very much for your feedback. The meaning of this section in the text precisely explains the meaning and importance of the formula. Revised lines 204-214 in the manuscript as follows:

“...Index meanings: 1. SHDI is the Shannon diversity index, which measures the complexity of the structure, Pi is the proportion of the total area of the landscape occupied by each patch type, m is the total number of patch types; 2. SHEI is the Shannon evenness index, H is the diversity index, Hmax is the maximum value of the diversity index; 3. Ci is the fragmentation index, Ni is the total number of patches in landscape type i, Ai indicates the area of landscape type i; 4. CONTAG is the spreading index, m is the total number of all patch types in the landscape; n is the number of patches in a given patch type, Pij is the probability that patch types i and j are adjacent; 5. FDI is the sub-dimensionality index, Pi is the perimeter of patch i, A is the area of patch i; 6. AI is the aggregation index, gii is the number of similar neighboring patches of the corresponding landscape type.”

Point 18: The land use transfer matrix is not described in the methodology section.

Response 18: Thank you for your hard work. The land use Stochastic matrix mainly relies on the analysis function of ArcGIS. In order to avoid the redundancy of the article, it has been omitted in this article.

Point 19: Lines 144 to 149, what is the meaning of this sentence?

Response 19: Thank you very much for your feedback. This sentence is mainly intended to illustrate the interpretation of rocky desertification through remote sensing images and other data. The specific steps can be found in the references.

Analysis of the results

Point 1: Lines 204 to 209: This sentence is very long, and the English writing is not smooth.

Response 1: Thank you very much for your feedback. We have rewritten the paragraph and have had it polished by a native English professional. Revised lines 217-224 in the manuscript as follows:

“..Figure 3 displays the spatial distribution of land use types in the Salaxi demonstration area. Cultivated land, shrubland, and woodland emerge as the primary land use types. In 2009 and 2019, the combined area of cultivated land, shrubland, and woodland accounted for 95.52% and 83.61% of the total area of the demonstration zone, respectively. Cultivated land is predominantly dry land, concentrated in relatively flat sections and non-karst plateau areas of Longfeng, Sala, Shale, Yongfeng, and other villages. Other land use types form linear strips on gentle slopes, alternating with shrubbery and woodland.”

Point 2: Lines 2010 to 211 “ The increased in shrub…..” This sentence is not clear.

Response 2: Thank you for your hard work. Thank you for letting us know where the foot is. We have thoroughly reviewed the manuscript and unified the names of the nine land use classifications. Revised lines 224-227 in the manuscript as follows:

“...Furthermore, there is a proportional increase in shrubland concurrent with a decrease in woodland. Grassland, in 2009, appears as patchy distributions in the northern and southern regions of the demonstration area. However, by 2019, ...”

Point 3: In Table 1. “what is the unit of the results?

Response 3: Thank you very much for your feedback. We have added units to Table 1. Revised lines 252 in the manuscript as follows:

“...Table 1. Land-use area change table of daozhen country from 2009 to 2019/ hm2

Point 4: The presentation in Table 1 is not enough. Make clear what is the gain or loss of different land use types.

Response 4: Thank you for your hard work. The land use Stochastic matrix has described various types of transfer areas. To avoid redundancy in the article, the percentage of transfer area was not increased.

Point 5: What is the classification method of the difference KD presented in Figure 2?

Response 5: Thank you for your hard work. The classification method of KD is also consistent with the method of land use. Both adopt the Spatial analysis Stochastic matrix method in ArcGIS.

Point 6: It will be interesting when you present the map related to the characteristics of the transition of KD and land use types.

Response 6: Thank you very much for your feedback. We have already conducted the image related to the transformation features of the two in section 3.3.2 of the article. Revised lines 345-360 in the manuscript as follows:

“3.3.2. Different KD levels and land use

...Figure 7 reveals that the KD landscape in the Salaxi demonstration area is primarily concentrated in cultivated land, shrubland, and forested land. In 2009, the area and proportion of mild and moderate KD in the same level were 892.61 hm2 (50.13%), 607.96 hm2 (34.14%), 211.40 hm2 (11.87%), and 370.20 hm2 (54.12%), 287.27 hm2 (42.00%), and 17.99 hm2 (2.63%), respectively. In 2019, the area and proportion of mild and moderate KD in the same level were 404.18 hm2 (43.16%), 359.35 hm2 (38.37%), 83.94 hm2 (8.96%), 27.66 hm2 (42.81%), 22.19 hm2 (34.35%), and 8.36 hm2 (12.93%), respectively. In 2009, the intensity of KD was mainly concentrated in shrubland and cultivated land, with an area and proportion of 117.35 hm2 (49.33%) and 108.96 hm2 (45.81%), respectively. In 2019, the intensity of KD was zero. Between 2009 and 2019, moderate and intense KD occurred in cultivated land, shrubland, Open woodland, forested land, and Unutilized land, showing a decreasing trend. However, mild and moderate KD occurred in grasslands, but the area did not increase significantly. This also indicates that the KD management projects such as returning farmland to forest and grass and returning to garden, and closing mountains for forestry have played a great role.”

Point 7: Line 239 what is the meaning of Salaxidemonstration?

Response 7: Thank you very much for your feedback. This is a language error we made during the writing process, and we have become aware of our own mistakes. We have made modifications to this. Revised lines 288-290 in the manuscript as follows:

“...From the trajectory map illustrating the evolution of KD in the Salaxi demonstration area (Fig. 5) and the data presented in Table 2,”

Point 8: Line 271, in the title of Figure 2. What is the meaning of RD?

Response 8: Thank you very much for your feedback. This is a language error we made during the writing process, RD should be changed to KD.

Point 9: Which methodology have you used to determine the results in Figures 4 and 5?

Response 9: Thank you for your hard work. We obtained the results based on the spatial stacking function in GIS. For example, first selecting the layer of non rocky desertification areas, and then cutting out various land use types, ultimately obtaining the land use classification of non rocky desertification areas.

Point 10: From lines 350 to 376, we account for only 2 sentences as long as this section is. Is it a serious

research paper? In addition, what is the value of this section? Results? Discussion?

Response 10: Thank you very much for your feedback. We have rewritten this paragraph to make it clearer for you and readers to understand. Revised lines 371-397 in the manuscript as follows:

“...The land use landscape index reveals a consistent increase in the Shannon Diversity index of land use within the Salaxi demonstration area throughout the study period, rising from 1.216 in 2009 to 1.480 in 2019. This trend signifies that the continuous progress of KD control project, Targeted Poverty Alleviation, industrial structure adjustment, and urbanization have contributed to an enhanced landscape diversity, heterogeneity, and complexity within the Salaxi demonstration area. Consequently, the differences in the areas of major land types have diminished. Furthermore, the Shannon evenness index also slightly increased, indicating that the evenness of various landscapes in the demonstration area has gradually increased, and the dominant land types have decreased, and the distribution of patches in each land type has become uniform. The fragmentation index and spread index reflect the connectivity, extension, and fragmentation degree of the landscape, with an increase in fragmentation index and a decrease in spread index. From 2009 to 2019, the fragmentation index of the Salaxi demonstration zone has been increasing year by year, indicating that with the construction of infrastructure, the previously larger patches have been cut into smaller ones. The fractal dimension index is close to 1 and has increased slightly in 2019 compared to 2009, indicating that the demonstration area is increasingly affected by human interference and the landscape shape is becoming more complex. The human restoration measures taken have improved the fractal dimension index of the landscape. The degree of convergence index decreased from 94.8652 in 2009 to 91.7795 in 2019. The KD project, the Grain for Green and grassland [38] project of returning fruits to garden, and infrastructure construction [39] caused large-scale landscape destruction and erosion, leading to increased landscape fragmentation, decreased spread, and poor landscape connectivity in the Salaxi demonstration area. In summary, human activities in the demonstration area have led to an increase in the fragmentation of land use landscapes, while also reducing the connectivity of the landscapes within the area. However, it may be beneficial for improving the ecological environment [40].”

Point 11: Globally, the results section is very long and confusing. It would be best if you rewrote it concisely and adequately.

Response 11: Thank you for your hard work. We have sorted out the analysis section in the article, hoping to make it clearer and clearer.

Discussions

Point 1: In the discussion section, the findings of this research need to be compared with similar work. So you can highlight your contribution.

Response 1: Thank you very much for your feedback. We have carefully considered your opinion. Summarized the previous achievements and compared them with this study.

Point 2: The paper has no discussion section to relate the findings to the theoretical statements.

Response 2: Thank you very much for your feedback. We have tried our best to link the research results with the theory in the article. Revised lines 438-453 in the manuscript as follows:

“...At the same time, there has been an increase in the diversity and heterogeneity of the land use landscape in the research area. The expansion of construction land, which includes urban residential, industrial, mining, and transportation areas, has resulted in the destruction of extensive areas of arable land, grassland, and forested landscapes. Consequently, this has led to an increase in landscape fragmentation, a decrease in spread, and a deterioration in landscape connectivity within the area. The diversity of KD landscape types has been reduced, and the intensity of KD is weakened, but the ecological restoration of the demonstration area makes the structure of KD landscape slightly complicated. Therefore, in the subsequent ecological management of KD and the project of returning farmland to forests and promoting fruit cultivation, while ensuring the increase of the area of characteristic economic forests and warp and fruit forests, it is crucial to consider the integrated impact of other human activities on the ecological landscape pattern. This approach aims to reduce disturbances to the ecological landscape pattern, promote human welfare, foster the healthy development of the ecosystem, and serve as a valuable reference for KD management in the southwest karst region.”

Point 3: It would help if you mentioned other studies, what results they got, and why you are getting better/different results.

Response 3:  Thank you very much for your feedback. We have carefully considered your opinion. Summarized the previous achievements and compared them with this study.

Point 4: The mechanism and the factors influencing the variability and Stability are not profoundly explained.

Response 4: Thank you for your hard work. We fully agree with your opinion, which is also the limitation of this study, as we have already explained in the discussion.

Point 5 and 6: Improve the discussion section, and consider the study's contribution, implication policies and research prospect. It seems there is no difference between the results and the discussion section.

Response 5 and 6: Thank you very much for your feedback. We strongly agree. At the same time, we made modifications in the text. Revised lines 429-437 in the manuscript as follows:

“...Regarding the higher incidence of KD in yellow soil and yellow-brown soil, it is recommended to select suitable ecological industries that align with the characteristics of these soil types. This approach would ensure efficient KD management while preserving soil quality and fertility. However, due to the limitations of the available data, the quantitative analysis conducted using two phases of images with time series compensation is insufficient to reveal the underlying mechanisms of these effects in depth. Therefore, further research is needed to explore the correlation characteristics between typical KD and land use, update the classification of KD, and elucidate the underlying mechanisms of these interactions.”

Conclusion

Point 1: In lines 377 to 380, the main objective of the research has not well explained, like in the induction section.

Response 1: Thank you very much for your feedback. We strongly agree. At the same time, we made modifications in the text. Revised lines 438-453 in the manuscript as follows:

“...At the same time, there has been an increase in the diversity and heterogeneity of the land use landscape in the research area. The expansion of construction land, which includes urban residential, industrial, mining, and transportation areas, has resulted in the destruction of extensive areas of arable land, grassland, and forested landscapes. Consequently, this has led to an increase in landscape fragmentation, a decrease in spread, and a deterioration in landscape connectivity within the area. The diversity of KD landscape types has been reduced, and the intensity of KD is weakened, but the ecological restoration of the demonstration area makes the structure of KD landscape slightly complicated. Therefore, in the subsequent ecological management of KD and the project of returning farmland to forests and promoting fruit cultivation, while ensuring the increase of the area of characteristic economic forests and warp and fruit forests, it is crucial to consider the integrated impact of other human activities on the ecological landscape pattern. This approach aims to reduce disturbances to the ecological landscape pattern, promote human welfare, foster the healthy development of the ecosystem, and serve as a valuable reference for KD management in the southwest karst region.”

Point 2: The conclusion section is very long. It must be summarized. Also, it's critical to highlight the limits of the study.

Response 2: Thank you very much for your feedback. We have refined certain statements in the conclusion section.

Point 3 and 4: In the conclusion section, the theoretical contribution of the study needs to be improved. The presentation of the conclusions is not suitable. Referring to the other research paper to concisely improve this section is suggested.

Response 3 and 4: Thank you very much for your feedback. We strongly agree. At the same time, we made modifications in the text. Revised lines 455-460 in the manuscript as follows:

“...This study integrates landscape pattern analysis with land use distribution pattern, KD distribution and evolution tracking, and soil types. The selected study area is the Salaxi Karst Plateau Mountainous KD Comprehensive Management Demonstration Area. The analysis examines the relationship between KD landscape patterns and spatial-temporal changes in land use type in the study area, yielding the following findings:”

Others remarks

Point 1: What is the theoretical contribution and significance of this paper?

Response 1: Thank you for your hard work. Exploring the response relationship between KD landscape and land use spatial-temporal change has important reference significance for the comprehensive control of KD in Karst Plateau plateau and its effects and regional sustainable development.

Point 2: What are the policies implications of the study?

Response 2: Thank you for your hard work. By studying the landscape patterns of KD and land use, reference can be provided for the later national strategies for KD control and the development of ecological industries.

Point 3: What is the innovation of the study?

Response 3: Thank you for your hard work. Thank you for your hard work. This study combines landscape pattern with land use distribution pattern, KD distribution and evolution track and Soil type, selects Salaxi Karst Plateau Plateau Mountainous KD Integrated Management Demonstration Area, and analyzes the response relationship between KD landscape pattern and spatial-temporal change of land use types in the study area. It has an important reference significance for the comprehensive control of KD in Karst Plateau plateau and its effects and regional sustainable development.

Point 4: English is not smooth and should improve it. You use long sentences that make your idea unclear.

Response 4: Thank you for your hard work. We have invited professionals whose native language is English to review the language of the manuscript.

Point 5: Studying the spatio-temporal land-use changes need at least two or three periods. Why do you only use a single period?

Response 5: Thank you for your hard work. We chose this period because it is a critical period for China's poverty alleviation efforts, with rapid development of agriculture, forestry, and animal husbandry. Choosing one stage is sufficient for the job, but choosing two would appear redundant, so we only used one stage.

Round 2

Reviewer 2 Report

We appreciate your effort to improve the quality of this paper. Unfortunately, this paper still has many problems that weaken the quality of your investigation.

Kindly see the report.

Best regards. 

The English language is not smooth and needs serious correction.

Author Response

Once again, we thank the reviewers for their hard work and we value their suggestions. And based on your feedback, we have made thorough revisions to the article. And the polishing agency under this MDPI has made English modifications, and the specific responses are described in the document.
